# Neurosymbolic Learning in Structured Probability Spaces: A Case Study

**Ole Fenske**                                            OLE.FENSKE@UNI-ROSTOCK.DE
**Sebastian Bader**                            SEBASTIAN.BADER@UNI-ROSTOCK.DE
**Thomas Kirste**                              THOMAS.KIRSTE@UNI-ROSTOCK.DE
*Hybrid Methods for Artificial Intelligence and Machine Learning, University of Rostock, Germany*

**Editors:** Leilani H. Gilpin, Eleonora Giunchiglia, Pascal Hitzler, and Emile van Krieken

## Abstract

This paper examines the impact of neurosymbolic learning on sequence analysis in Structured Probability Spaces (SPS), comparing its effectiveness against a purely neural approach. Sequence analysis in SPS is challenging due to the combinatorial explosion of states and the difficulty of obtaining sufficient annotated training samples. Additionally, in SPS, the set of realizations with non-zero support is often a scattered, non-trivial subset of the Cartesian product of variables, adding complexity to learning and inference. The problem of sequence analysis in SPS emerges, for example, in reconstructing the activities of goal-directed agents from noisy and ambiguous sensor data. We explore the potential of neurosymbolic methods, which integrate symbolic background knowledge with neural learning, to constrain the hypothesis space and improve learning efficiency. Specifically, we conduct a simulation study in human activity recognition using DeepProbLog as a representative for neurosymbolic learning. Our results demonstrate that incorporating symbolic knowledge improves sample efficiency, generalization, and zero-shot learning, compared to a purely neural approach. Furthermore, we show that neurosymbolic models maintain robust performance under data scarcity while offering enhanced interpretability and stability. These findings suggest that neurosymbolic learning provides a promising foundation for sequence analysis in complex, structured domains, where purely neural approaches struggle with insufficient training data and limited generalization ability.

## 1. Introduction

In this paper, we compare the efficiency of neurosymbolic vs. purely neural sequence modeling in Structured Probability Spaces (SPS). Such spaces often contain combinatorial or highly structured objects, making probabilistic filtering challenging due to complex dynamics and high space complexity. A notable example is Human Activity Recognition (HAR), where internal structure and temporal dependencies increase difficulty. While purely neural methods have shown success in HAR (Minh Dang et al., 2020; Chen et al., 2021), they typically require extensive labeled data—problematic in applications with limited annotation. To mitigate data scarcity, domain-specific background knowledge can be leveraged. Past work (Krüger et al., 2013, 2014) has used symbolic knowledge for inference in SPS, yet these methods rarely integrate deep learning components that excel at modeling complex observations. DEEPPROBLOG (Manhaeve et al., 2018) now addresses this gap, combining neural and symbolic-probabilistic approaches.

Our study uses DEEPPROBLOG (and a purely neural baseline) to explore three core learning abilities important for SPS: Sample Efficiency, Generalizability and Zero-Shot Learning.

The remainder of this paper is organized as follows: Section 2 outlines learning in SPS. Section 3 introduces DeepProbLog as a framework for neurosymbolic AI. Section 4 presents our application scenario and method. Section 5 discusses the experiments and findings. Finally, Section 6 concludes and proposes future work.

## 2. Learning in Structured Probability Spaces

Many real-world systems—ranging from route planning to goal-directed human activities—exhibit structurally constrained dynamics. In such Structured Probability Spaces (SPS), valid state transitions follow specific rules. This means that valid state sequences are a small scattered subset of the Cartesian product over all sequences of state variables. Therefore, learning in such spaces is challenging due to (1) the exponential growth of potential state sequences and (2) the non-availability of annotated data on infeasible events, which cannot occur in practice.

### 2.1. Logic vs. Deep Learning

**Logic-based approaches**, especially those integrating probabilistic reasoning (e.g., PSDDs (Kisa et al., 2014), ProbLog (De Raedt et al., 2007) or CCBM (Krüger, 2016)), leverage domain knowledge to limit the hypothesis space, leading to more efficient inference. For instance, Krüger et al. (2014) showed that such an approach can scale to large state spaces, containing more than $10^8$ states. Their CCBM system (Krüger, 2016) uses the Planning Domain Definition Language to constrain the search space to valid states only. Such methods offer efficient inference (constraining valid events reduces computational overhead) and enhanced interpretability (explicit domain knowledge illuminates system behavior). However, they typically require detailed manual modeling and cannot automatically learn features from raw sensor data.

**Neural network models** (e.g., CNNs, RNNs) in contrast, excel at extracting features from raw sensor data (Chen et al., 2021; Wan et al., 2020), showing success in recognizing activities from noisy sensor data. Unified models combine multiple neural architectures to classify simple and complex activities simultaneously (Huan et al., 2022; Mekruksavanich et al., 2022a,b; Bouton-Bessac et al., 2023), whereas separated models first detect simpler actions and then infer more complex activities (Peng et al., 2018; Cheng et al., 2018; Chen et al., 2023) from the simpler ones. While these methods automate feature learning, they rely on large datasets, which is often impractical in domains with annotation scarcity. Furthermore, they ignore domain constraints, thus failing to exploit structural knowledge which can hinder generalization and interpretability of such systems.

### 2.2. Motivation for a Neurosymbolic Approach

As can be seen, both approaches (Logic and Deep Learning) complement each other in their strengths and weaknesses. Neurosymbolic (NeSy) AI merges these paradigms, aiming to: (1) Improve sample efficiency by incorporating domain rules, (2) enhance generalization via structural constraints and (3) increase robustness and interpretability, bridging the gap between explicit reasoning and automatic feature learning.

These claims are also supported by Darwiche (2016), who frames such an integration as

"learning from data and knowledge". Additionally, he argues that logic can also be used for factoring the respective structured probability space into a tractable representation, allowing not only learning but also reasoning in a more efficient way. This synergy promises more robust learning in SPS, prompting our comparative investigation of neurosymbolic methods versus purely neural baselines.

## 3. DeepProbLog in a Nutshell

DEEPPROBLOG (Manhaeve et al., 2018) is a framework for NeSy-AI and extends the probabilistic logic language PROBLOG (De Raedt et al., 2007) by integrating neural predicates directly into the PROBLOG language. This allows neural networks to provide probability estimates for specific logical facts, merging data-driven feature learning with probabilistic symbolic reasoning.

### 3.1. From Prolog to ProbLog

PROLOG is a logic programming language that uses facts and rules (e.g., `a.  b :- a.`) to determine whether a query (e.g., `b.`) follows from a program. PROBLOG adds probabilistic annotations to PROLOG facts, inducing a probability distribution over all "deterministic" programs. Consider the PROBLOG program $L =$ "`0.7::a.  b:-a.`" This represents the idea that the fact `a` is contained with probability 0.7 in the program. The two resulting *deterministic* programs, are given by `a`: $L_{\{a\}} =$ "`a. b:-a.`" and `a`: $L_{\{\}} =$ "`b:-a.`". The probability of a query for the given program being true is then computed by summing over all deterministic programs where the query holds, weighted by their respective probabilities. A more detailed explanation about the syntax and semantics of PROBLOG is provided in Appendix A.

### 3.2. Neural Predicates in DeepProbLog

DEEPPROBLOG extends PROBLOG by connecting a neural network's outputs to probabilistic facts in a PROBLOG program. Consider a classification problem where $\mathcal{O}$ is a set of observations, $\mathcal{X} = \{x_1, \ldots, x_C\}$ a set of $C$ class labels, and $P(x \mid o)$ the probability that observation $o \in \mathcal{O}$ is of class $x \in \mathcal{X}$. For a vector of observations $\mathbf{o} = o_{1:T}$, $o_t \in \mathcal{O}$, we could take the probabilities $\theta_{tc} = P(x_c \mid o_t)$ and then use these probabilities (which sum to one for a given $t$) to define a so-called annotated disjunction (AD) for each $t$, given by $\theta_{t1}$::`class(`$t$`,`$x_1$`);` $\ldots$ $\theta_{tC}$::`class(`$t$`,`$x_C$`)`. Such an AD declares that the $C$ facts are mutually exclusive; exactly one of them is contained in a deterministic program. The fact `class(`$t$`,`$x_c$`)` states that observation $o_t$ is of class $x_c$. The value $\theta_{tc}$ is the probability that this fact holds. Computing the values $\theta_{tc}$ can in principle be performed in a preprocessing step, which generates a list of $T$ annotated disjunctions (with $C$ elements each) that are added to the rest of the PROBLOG program prior to further processing. (Indeed, the values $\theta_{tc}$ can be pictured as a $T$ by $C$ matrix, one row for each observation, one column for each class label.)

This concept is implemented in DEEPPROBLOG by so-called neural annotated disjunctions (nADs). Assume there is a function $O(\boldsymbol{o})$, given by a neural network, that produces a parameter matrix $(\theta_{tc})$ from an observation vector $\boldsymbol{o}$. A nAD using $O$ is then declared by:

$$nn(O,\texttt{[Ot]},\texttt{Xc},[x_1,\ldots,x_C])\texttt{::class(Ot,Xc)}.$$

The list $[x_1,\ldots,x_C]$ labels the columns of the $\theta_{tc}$ matrix. This nAD is then replaced by its respective AD when DEEPPROBLOG grounds a program with respect to a query. In this way DEEPPROBLOG seamlessly embeds neural predictions (e.g., from CNNs or RNNs) into a larger probabilistic-symbolic model, enabling end-to-end neurosymbolic inference and learning. For further details about learning in DEEPPROBLOG see Appendix B.

## 4. Method

We demonstrate how DEEPPROBLOG can leverage symbolic domain knowledge alongside neural inference. After outlining a simplified indoor activity scenario, we show how to encode it within DEEPPROBLOG and discuss the resulting probabilistic model.

### 4.1. Example domain

We adapt the scenario from Krüger et al. (2012), where a single person performs tasks in a small room (see Figure 1). The user's goals—printing documents and making coffee—must follow certain constraints (e.g., carrying only one item at a time, ensuring that printer and coffee machine are refilled with paper or water). The state of the system is defined by multiple state variables:

- Location L∈{door,paper_stack,printer,water_tap,coffee_machine}

- Printed status P∈{printed,notPrinted}

- what the user holds H∈{nothing,paper,water,coffee}

- Status of printer PP∈{paper,noPaper}

- Status of coffee machine MW∈{water,noWater}

The user can perform several actions: go to another location, fetching paper or water, replenish paper or water, making coffee or simply doing nothing. In total the user can apply 12 actions (5 actions for going to another location and the 7 actions that can be performed at the different locations). For each state of the system, the user can apply exactly 5 of these 12 actions (go to the other 4 locations or 1 of the location dependent actions). Moreover, we simulate low-resolution sensor data (1-channel 8x8 thermal images) to indicate the person's position, focusing on how to robustly model action sequences rather than handling high-dimensional inputs. An example of such sensor data can be seen in Figure 2.

### 4.2. Implementation in DeepProbLog

To encode this domain, we define a `state(L,P,H,PP,MW)` predicate which variables are equivalent to the state variables already defined for our domain. The actions in our domain can be described by precondition-effect rules used in the Planning Domain Definition Language (PDDL). The `print` action for example is defined in PDDL as follows:

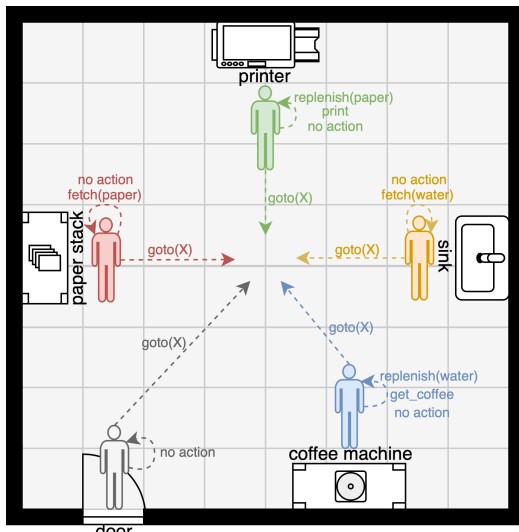

Figure 1: Floor plan of the room with locations and applicable actions.

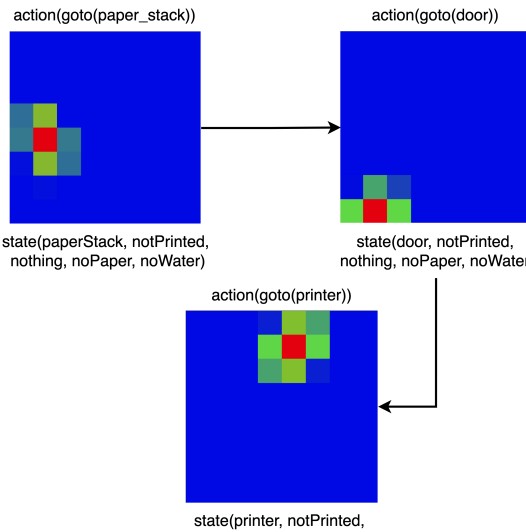

Figure 2: A data sample with three thermal images.

```
(: action print
    : precondition (and
        (not printed)
        (has printer paper)
        (at printer))
    : effect (printed))
```

To implement such a precondition-effect rule in DᴇᴇᴘPʀᴏʙLᴏɢ we use a simple predicate `action(AT,ST1,ST)`, where `AT` describes the name of the corresponding action, `ST1` defines how the world has to look like to apply the action `AT` (=preconditions) and `ST` describes how the world changes after applying `AT` (=effect). For our `print` action the corresponding predicate in DᴇᴇᴘPʀᴏʙLᴏɢ looks like the following:

```
action(print ,
    state(printer ,notPrinted ,H, hasPaper ,MW) ,
    state(printer ,  printed ,   H, hasPaper ,MW)).
```

In general the `action` predicates are deterministic. This means that we can apply exactly one action for every situation (combination of old state `ST1` and new state `ST`) we can encounter. In contrast, the observations (in our case location `L`) we make are probabilistic in nature and draw their distribution directly from a concrete sensor measurement $y_t$. For this purpose we use a neural annotated disjunction to implement a predicate that defines a probability distributions over possible observations in DᴇᴇᴘPʀᴏʙLᴏɢ:

```
nn(net ,YT,L,[ door , paperStack , printer ,waterTap , coffeeMachine ])  ::  obs(T,YT,L).
```

As it can be seen, the nAD maps a thermal image `YT` to a location `L`. A simple CNN `net` (detailed in Subsection 5.2) outputs the probabilities for the labels of `L`, which DᴇᴇᴘPʀᴏʙLᴏɢ then treats as the probabilities of the corresponding facts `obs(T,YT,L)`.

To compute the probabilities over final states $P(S_T|y_{1:T}, s_0)$ we use the following recursive rule:

```
filter(0,[],state(door,notPrinted,nothing,noPaper,noWater)).

filter(T,[YT|YS],ST):-
    T1 is T-1,
    filter(T1,YS,ST1),
    action(AT,ST1,ST),
    ST == state(L,_,_,_,_),
    obs(T,YT,L).
```

The `filter(0,[],...)` predicate defines the initial state $s_0$ of the system. The recursive rule `filter(T,[YT|YS],ST)` is then just unrolled and aligns the single actions `AT` with the observed location `L` in `obs(T,YT,L)` by enforcing equality of the location in state `ST` (which results from applying `AT` for state ST1) with the observed location. For additional details what kind of probabilistic model results from this approach, please see Appendix C.

## 5. Evaluation

We designed three experiments to compare DeepProbLog with a purely neural CNN-RNN baseline in our Structured Probability Space (SPS) domain. Each experiment highlights a different learning challenge: sample efficiency, generalization, and zero-shot learning.

### 5.1. Task and Hypotheses

All experiments use a 3-step prediction task. We start from a known state $s_0$ (as given by `filter(0,[],state(door,notPrinted,nothing,noPaper,noWater)))`. As already mentioned in this model, each state allows a subset of five actions. After three steps, this yields 125 possible action sequences which can result in 15 possible final states. Depending on the experiment we either want to compute the distribution over final states or over possible action sequences.

We examine three hypotheses tied to key learning properties:

1. H1 (Sample Efficiency): Adding symbolic constraints reduces the data required to achieve a given performance level.

2. H2 (Generalizability): When symbolic knowledge is present, the system is still able to correctly recognize final states for which a certain amount of action sequences have been removed from train data.

3. H3 (Zero-Shot Learning): The model recognizes final states that never appeared in training, provided it has seen training samples for all possible observations and has symbolic knowledge.

### 5.2. Experimental Setup

**Data.** We generated 3750 synthetic training samples for each experiment. As we have 125 possible action sequences and 15 final states this equals to 30 samples for each sequence or 250 samples for each final state (depending on the experiment). Each training sample

consists of three thermal images and is associated with the states and the underlying action sequence (as can be seen in Figure 2). Depending on the hypothesis tested, we remove/withhold certain sequences or states from the training set in a controlled manner. The test set is generated independently for each experiment and contains in total 750 samples (6 samples for each action sequence).

**Models.** Our DeepProbLog model uses a CNN which processes each thermal image, yielding probabilities over five possible locations (`door`, `printer`, etc.). The symbolic part then uses logical predicates to constrain which actions and state transitions are valid, effectively filtering out impossible sequences. The final state distribution is computed by unrolling these transitions over three time steps. The CNN-RNN baseline model uses a slightly different CNN that extracts a latent representation from each image. An RNN then models sequential dependencies directly in a purely neural way, predicting the final state distribution after three steps. This setup allows direct comparison of how symbolic knowledge affects performance on various data reduction scenarios. A more detailed description of the used neural networks can be found in Appendix D.

**Hyperparameters.** For training both models we use a learning rate of 0.001 and early stopping with a patience of 4. The model training for a respective train set is repeated five times by using different seeds for initializing its weights before training. The performance after a single training session is measured by the Macro-F1-Score on the respective test set. As we execute training multiple times with different initial weights, we then take the mean of the resulting Macro-F1-Scores and also compute its standard deviation.

### 5.3. Experiment 1: Sample Efficiency

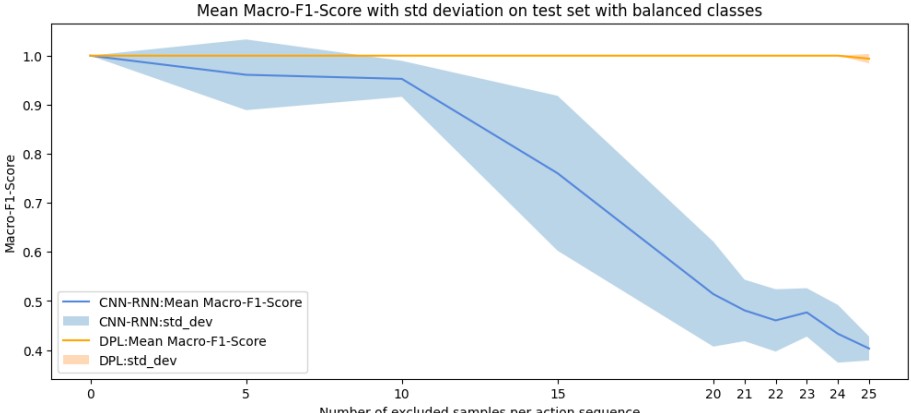

Figure 3: Results of experiment 1 using our DeepProbLog (DPL) approach and a CNN-RNN baseline model.

Objective of this experiment is to test H1, which concerns the effect of background knowledge in sample efficiency. Specifically, we assume that the neurosymbolic model requires less training data than the neural model in order to reach optimal performance. To test this, we remove increasing amounts of training data and monitor the resulting model performance.

**Procedure.** We start with a train set with 30 samples for each action sequence. We removed 5 samples at a time until 10 only remain. Afterwards we decrease the number of sample for each action sequence by 1 until only 5 remain. Both models are retrained and tested for each of the reduced train datasets. The test set remains untouched.

**Results.** In Figure 3 we can see the results for both approaches. The x-axis shows how many samples for each sequence are excluded from the train set and the y-axis displays the Mean Macro-F1-Score on the test set. As it can be seen DEEPPROBLOG maintains a high Macro-F1-Score even with 25 of 30 training samples removed for each action sequence (which equals 16.6% of the initial train data set). The CNN-RNN's Macro-F1-Scores start to drop notably as only 50% of all training samples are removed. Thus, H1 is supported: symbolic constraints help preserve performance under limited data. Moreover, the standard deviations show that DEEPPROBLOG is more stable across random weight initializations, suggesting that the addition of symbolic knowledge not only aids sample efficiency, but also stability of the model.

### 5.4. Experiment 2: Generalizability

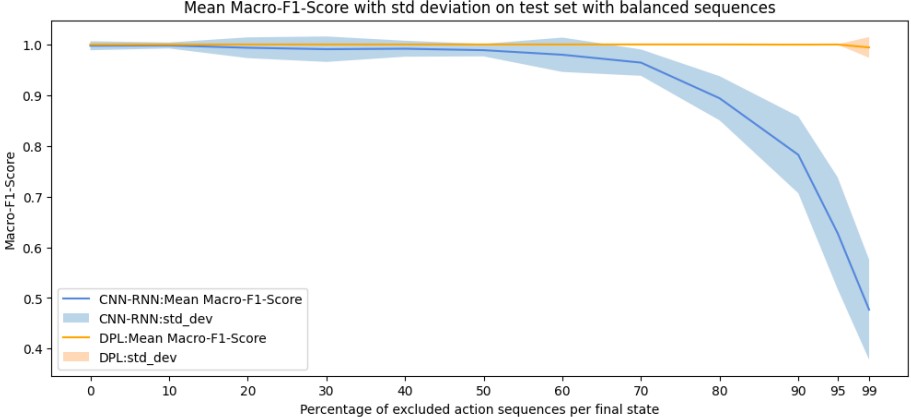

Figure 4: Results of experiment 2 using our DEEPPROBLOG (DPL) approach and a CNN-RNN baseline model.

Objective of this experiment is to test H2, concerning the generalization ability. Specifically, we assume that the neurosymbolic model is better able to correctly recognize final states for which a certain amount of action sequences have been removed from train data. To test this, we remove an increasing amount of sequences from the training data for each final state and monitor model performance on all final states.

**Procedure.** As in experiment 1 we start with the complete train data. We increase the percentage of withhold action sequences for each final state from the train data by 10% until we removed 90% of all sequences. Afterwards, we delete 95% and 99% of all sequences. In this experiment we additionally vary the seeds used for removing a certain percentage of sequences to eliminate the possibility of good or bad performance due to a bad deletion order. As in experiment 1 both models are retrained and tested for each of the reduced

train datasets and the test set remains untouched.

**Results.** In Figure 4 we can see the results for this experiment. DeepProbLog consistently performed well, even with 99% of all sequences removed from the train set. The performance of the CNN-RNN approach starts to degrade as 50% of all sequences are removed. It even becomes worse when 70% of all sequences are deleted from the training data. Based on these results we can confirm H2: symbolic domain knowledge aids in inferring valid transitions despite missing entire action sequences. Moreover, variance analyses indicated that the CNN-RNN's initial parameter seed had a large effect on final performance, whereas DeepProbLog was not sensitive to its initial weights, thus supporting the same claim already made in Experiment 1.

### 5.5. Experiment 3: Zero-Shot learning

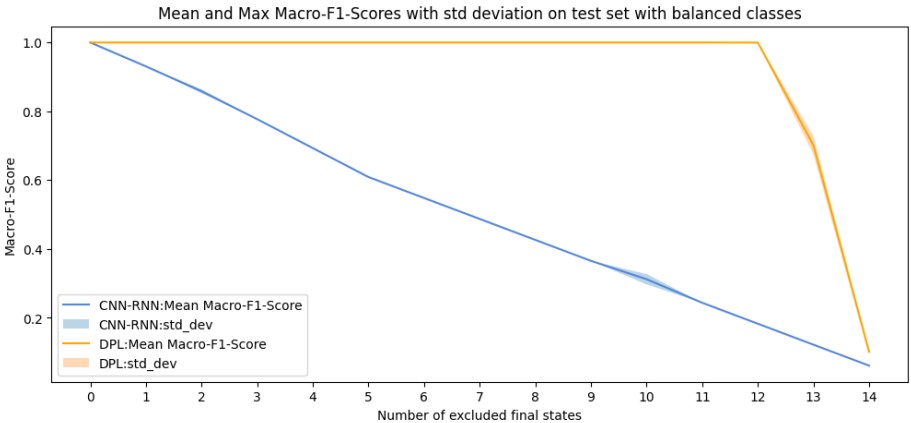

Figure 5: Results of experiment 3 using our DeepProbLog (DPL) approach and a CNN-RNN baseline model.

The third experiment tests hypothesis H3. In this context we want to analyse how well a model can recognize final states that never appeared in the training data. To this end we systematically remove complete final states from the training set. Through this procedure, we can analyze the zero-shot learning abilities of both models for previously unseen final states.

**Procedure.** We start with a training data set that consists of 15 final states with 250 samples for each state. We then remove one final state (with all of their samples) after the other, until only 1 final state remains in the train set. Both models are then retrained for each of the reduced train sets and tested on a separate test set which still contains samples for all valid final states and remains untouched throughout the whole experiment.

**Results.** In Figure 5 we can see the overall results for this experiment. DeepProbLog can successfully identify unseen final states, provided all relevant location observations appear somewhere in the training data. DeepProbLog 's accuracy starts to decrease when only 2 final states are left. This happens because the two final states do no longer contain sensor data for all 5 location observations we can make. Nevertheless, the CNN-RNN struggled

right from the beginning and its performance decreases linear with the amount of removed final states, likely because it attempts to learn a direct mapping from sensor data sequences to final states without paying attention to the logical constraints inherent in the domain. Overall, we conclude H3 holds, especially when training data retain coverage of all possible observations.

### 5.6. Summary and Limitations

From the experiments we can conclude that neurosymbolic learning in general is well suited for learning within SPS. The symbolic knowledge helps in limiting the hypotheses space to the space of possible events only and therefore can increase sample efficiency as shown with experiment 1. Moreover, the use of symbolic knowledge helps with generalizing to previously unseen action sequences and states by allowing the learner to focus on extracting observations from the raw sensor data rather than also learning the constraints inherent in an SPS. Experiment 2 and 3 have shown this.

Nevertheless, our studies also have some limitations. First, we used a fairly simple and small domain which only consists of 125 states and 12 actions, as it is well known that DeepProbLog (as many other NeSy approaches) has scalability issues (it has to unroll the respective SPS completely in time to process a query). However, this issue is addressed by the recently published Relational Neurosymbolic Markov Models (NeSy-MMs) (Smet et al., 2024), which marginalize over previous time steps. Therefore, future work could apply this technique to more complex scenarios, such as the one used by Krüger et al. (2014), to see how well NeSy-AI can already scale to real-world scenarios. Second, the modeling task itself can also quickly become more complicated (and therefore error-prone) when analyzing such bigger domains. This calls for methods that can automatically extract such models from, for example, textual data (Stoev et al., 2023), which could be combined with NeSy-AI approaches. Last, our study only includes DeepProbLog as a representative for neurosymbolic learning, as we want to show how NeSy-AI can tackle the different challenges related to learning within SPS. Although, comparing a broader range of NeSy system (e. g. DeepSeaProbLog (Smet et al., 2023), Logic Neural Networks (Riegel et al., 2020), Logic Tensor Networks (Badreddine et al., 2022), etc.) against each other for different problem setups (e.g. discrete-continuous, sequential, SPS, etc.) could highlight new insights regarding which approach is best suited for what kind of environment/task.

## 6. Conclusion

In this paper, we explored the benefits of neurosymbolic learning in SPS. By using DeepProbLog as a representative for neurosymbolic learning, we have shown how such a system can be used for the task of sequential state estimation from noisy sensor data. Moreover, through controlled experiments, we evaluated multiple learning hypotheses and compared the NeSy approach with a purely neural baseline, highlighting the advantages of incorporating symbolic knowledge into the learning process and inference. Future work could address the points outlined in Subsection 5.6, such as testing more sophisticated approaches for larger domains or extracting symbolic knowledge automatically from data. Overall, this paper contributes towards the direction of using NeSy-AI for inference and learning in SPS, thus paving the way for more intelligible recognition systems.

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

## Appendix A. Probabilistic Syntax and Semantics of ProbLog

PROBLOG extends PROLOG syntax by allowing to annotate facts with probabilities. Consider the PROBLOG program $L =$ "`0.7::a.  b:-a.`" This represents the idea that the fact `a` is contained with probability 0.7 in the program. Such a program is called a *probabilistic logic program*. The semantics of the probabilistic program $L$ is a probability distribution over queries. This can be explained as follows: PROBLOG (conceptually) creates two *deterministic* programs, one including the probabilistic fact `a`: $L_{\{a\}} =$ "`a. b:-a.`" and one without `a`: $L_{\{\}} =$ "`b:-a.`". The probabilities for these programs are given by the probability annotations of $L$: $P(L_{\{a\}}) = 0.7$ and $P(L_{\{\}}) = 0.3$. The probability that a query $q$ is a logical consequence of $L$ is then simply given by $P(L \models q) = 0.7 \cdot [L_{\{a\}} \models q] + 0.3 \cdot [L_{\{\}} \models q]$. For instance, $P(\text{"}b.\text{"}) = 0.7 \cdot [L_{\{a\}} \models \text{"}b.\text{"}] + 0.3 \cdot [L_{\{\}} \models \text{"}b.\text{"}] = 0.7 \cdot 1 + 0.3 \cdot 0 = 0.7$.

In general, let $L$ be a PROBLOG program with $K$ annotated facts $\theta_k \! : \! : \! \alpha_k$, with $\mathcal{A} = \{\alpha_1, \ldots \alpha_K\}$ being the set of all such facts. Let $\boldsymbol{\alpha} \subseteq \mathcal{A}$, be a subset of these facts. Then the probability of this choice is given by $\pi_{\boldsymbol{\alpha}} = \prod_{k=1}^{K} \theta_k^{[\alpha_k \in \boldsymbol{\alpha}]} \cdot (1 - \theta_k)^{[\alpha_k \notin \boldsymbol{\alpha}]}$. The $2^K$ choices $\boldsymbol{\alpha} \subseteq \mathcal{A}$ enumerate all deterministic programs $L_{\boldsymbol{\alpha}}$ that can be generated from $L$. By construction, $\sum_{\boldsymbol{\alpha} \subseteq \mathcal{A}} \pi_{\boldsymbol{\alpha}} = 1$. So, $\pi$ is a sound distribution over the $L_{\boldsymbol{\alpha}}$. If we consider "$L \models q$" as a Boolean random variable and $L$ as a random variable with realizations $L_{\boldsymbol{\alpha}}$, we obtain:

$$
\begin{aligned}
P(L \models q) &= \sum_{\boldsymbol{\alpha} \subseteq \mathcal{A}} P(L \models q, L{=}L_{\boldsymbol{\alpha}}) \\
&= \sum_{\boldsymbol{\alpha} \subseteq \mathcal{A}} P(L \models q \,|\, L{=}L_{\boldsymbol{\alpha}}) \cdot P(L{=}L_{\boldsymbol{\alpha}}) \\
&= \sum_{\boldsymbol{\alpha} \subseteq \mathcal{A}} [L_{\boldsymbol{\alpha}} \models q] \cdot \pi_{\boldsymbol{\alpha}}.
\end{aligned}
\tag{1}
$$

Eq. (1) suggests the following procedure. Using a suitable inference method, such as PROLOG, we compute $[L_{\boldsymbol{\alpha}} \models q]$ for all realizations $L_{\boldsymbol{\alpha}}$ of $L$ and sum the weighted results. However, this procedure clearly is very inefficient. To solve this, PROBLOG transforms (1) into an arithmetic circuit that provides efficient evaluation (see (De Raedt et al., 2007)).

## Appendix B. Parameter learning in DeepProbLog

In a probabilistic logic program with $K$ parameters $\boldsymbol{\theta} = \theta_{1:K}$, it may be of interest to estimate these parameters from training data. In DEEPPROBLOG, gradient descent is used. We here discuss, how this is realized.

### B.1. Finding the Objective

Eq. (1) introduced $P(L \models q)$, the function defined by a PROBLOG program $L$ for a query $q$. We now write this as $P_q(\boldsymbol{\theta})$ to make explicit its dependence on the parameters. In the previous section, we have introduced the parameter computation $O(\mathbf{o})$, which itself may also depend on a parameter vector $\boldsymbol{\phi}$. Here, we also make the parameter dependence explicit by writing $O_{\mathbf{o}}(\boldsymbol{\phi})$.

To simplify things, let us assume that $O_{\mathbf{o}}(\boldsymbol{\phi})$ produces *all* parameters required by $P_q$ (maybe simply by passing some values of $\boldsymbol{\phi}$ unchanged to its output). Parameter estimation thus can focus on $\boldsymbol{\phi}$.

For a given query $q$ with associated observation vector $\mathbf{o}_q$, DEEPPROBLOG uses $\boldsymbol{\phi}$ to compute $\hat{u}_q$, an estimate of the true probability $u_q$ of $q$ being a logical consequence of $L$, by

$$\hat{u}_q = (P_q \circ O_{\mathbf{o}_q})(\boldsymbol{\phi}) \tag{2}$$

### B.2. The Loss Function

The probability that $q$ is the logical consequence of a random program sampled from $L$ is described by a Bernoulli random variable with distribution parameter $u_q$. The value $\hat{u}_q$ approximates $u_q$. If $u_q$ is the training target, we can use the objective of minimizing the Kullback-Leibler divergence between the target distribution defined by $u_q$ and the estimated distribution given by $\hat{u}_q$. This is achieved by minimizing the cross entropy. For two Bernoulli random variables with parameters $u$ and $\hat{u}$, this is $h(u, \hat{u}) = u \cdot \log \hat{u} + (1 - u) \cdot \log(1 - \hat{u})$. Writing this as $h_u(\hat{u})$ and combining it with (2) gives the complete objective:

$$J_{q,u_q,\mathbf{o}_q}(\boldsymbol{\phi}) = (\overbrace{h_{u_q} \circ P_q}^{\text{loss}} \circ O_{\mathbf{o}_q})(\boldsymbol{\phi}) \tag{3}$$

The composition $h_{u_q} \circ P_q$ contains the computation of the estimated probability of $q$ being a logical consequence and the comparison of this estimate to a target probability via cross entropy. From the viewpoint of the neural network $O$, this composite simply constitutes the loss function.

### B.3. The Training Loss

Given a sequence of $N$ training triples $(q_n, \mathbf{o}_n, u_n)$, we minimize the average loss:

$$J(\boldsymbol{\phi}) = \frac{1}{N} \sum_{n=1}^{N} (\overbrace{h_{u_n} \circ P_{q_n}}^{\text{loss}_n} \circ O_{\mathbf{o}_n})(\boldsymbol{\phi}) \tag{4}$$

Often, the target values $u_n$ are simply 1 or 0, stating that $q_n$ has found to be true (or false) in training data collection.

### B.4. The Jacobian of $J(\phi)$

The Jacobian of an objective function evaluated at the current parameter values is the basis for parameter estimation by gradient descent. By the chain rule (and linearity of differentiation), the Jacobian of $J$ is

$$\mathbf{J}_J(\boldsymbol{\phi}) = \frac{1}{N} \sum_{n=1}^{N} \underbrace{\overbrace{\mathbf{J}_{h_n}(\hat{u}_n) \cdot \mathbf{J}_{P_n}(\boldsymbol{\theta}_n)}^{\mathbf{J}_{\text{loss}_n}(\boldsymbol{\theta}_n)} \cdot \mathbf{J}_{O_n}(\boldsymbol{\phi})}_{\mathbf{J}_n(\boldsymbol{\phi})} \tag{5}$$

where $\mathbf{J}_{h_n}$ is the Jacobian of $h_{u_n}$ (which is a simple scalar), $\mathbf{J}_{P_n}$ the Jacobian of $P_{q_n}$, and $\mathbf{J}_{O_n}$ the Jacobian of $O_{\mathbf{o}_n}$, with $\boldsymbol{\theta}_n = O_{\mathbf{o}_n}(\boldsymbol{\phi})$ and $\hat{u}_n = P_{q_n}(\boldsymbol{\theta}_n)$.

### B.5. Performing Training

$\mathbf{J}_J(\phi)$ is computed as follows: For all samples $(q_n, \mathbf{o}_n, u_n)$: (1) Compute $\boldsymbol{\theta}_n = O_{\mathbf{o}_n}(\phi)$ by the neural network (which also prepares for computing $\mathbf{J}_{O_n}(\phi)$ by backward-mode autograd). (2) Compute $\hat{u}_n = P_{q_n}(\boldsymbol{\theta}_n)$ using the arithmetic circuit, giving also $\mathbf{J}_{P_n}(\boldsymbol{\theta}_n)$ by forward-mode autograd. (3) Compute cross entropy $h_{u_n}(\hat{u}_n)$ and Jacobian $\mathbf{J}_{h_n}(\hat{u}_n) = \frac{u_n}{\hat{u}_n} - \frac{1-u_n}{1-\hat{u}_n}$. (4) Compute loss Jacobian $\mathbf{J}_{\mathrm{loss}_n}(\boldsymbol{\theta}_n) = \mathbf{J}_{h_n}(\hat{u}_n) \cdot \mathbf{J}_{P_n}(\boldsymbol{\theta}_n)$. (5) Push $\mathbf{J}_{\mathrm{loss}_n}(\boldsymbol{\theta}_n)$ back into the neural network to get $\mathbf{J}_n(\phi)$. (In PyTorch, this is done by `torch.autograd.backward(`$\boldsymbol{\theta}_n$`,grad_tensors=` $\mathbf{J}_{\mathrm{loss}_n}(\boldsymbol{\theta}_n)$`)`.) (6) Sum all $\mathbf{J}_n(\phi)$ and divide by $N$. The whole training process can be also seen in Figure 6.

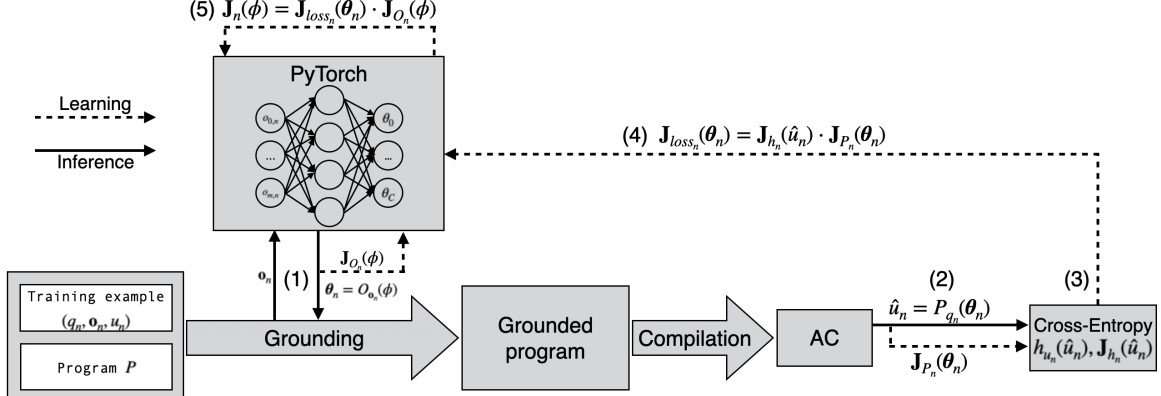

Figure 6: The inference and learning process in DeepProbLog.

## Appendix C. Probabilistic model of our approach

A Maximum-Entropy-Markov-Model is a simple Maximum-Entropy classifier (e. g. multivariate logistic regression) but adds additional dependencies/transitions between the latent variables we want to predict. Therefore we assume that the unknown values $X_t$ we want to predict are organized in a Markov chain rather then being conditionally independent from each other. This allows us to model the temporal dependencies, like in a HMM, but reversing the causal relation between $X_t$ and $y_t$. The computation of $P(X_T|y_{0:T})$ can now be done in the following way:

$$
\begin{aligned}
P(X_T|y_{0:T}) &= \frac{P(X_T, y_{0:T})}{P(y_{0:T})} \\
&= \frac{\sum_{X_{T-1}} P(X_T, X_{T-1}, y_{0:T})}{P(y_{0:T})} \\
&= \sum_{X_{T-1}} P(X_T|X_{T-1}, y_T) P(X_{T-1}|y_{0:T-1}) \\
&= \sum_{X_{0:T-1}} \prod_{t=0}^{T} P(X_t|X_{t-1}, y_t)
\end{aligned}
$$

Our approach further extends this standard MEMM such that $X_t$ itself is a Structured Probability Space, meaning that it is composed of hidden random variables for the state $S_t$, the applied action $A_t$ and the observation $O_t$, which have several dependencies and constraints. The state $S_t$ for example depends on the applied action $A_t$ and the former state $S_{t-1}$. The action $A_t$ whereas is dependent on the observation $O_t$ and the former state $S_{t-1}$. The observation itself only depends on $y_t$ (which is in our scenario a thermal image) and has no other temporal dependencies. The resulting model can be seen at the right in Figure 7.

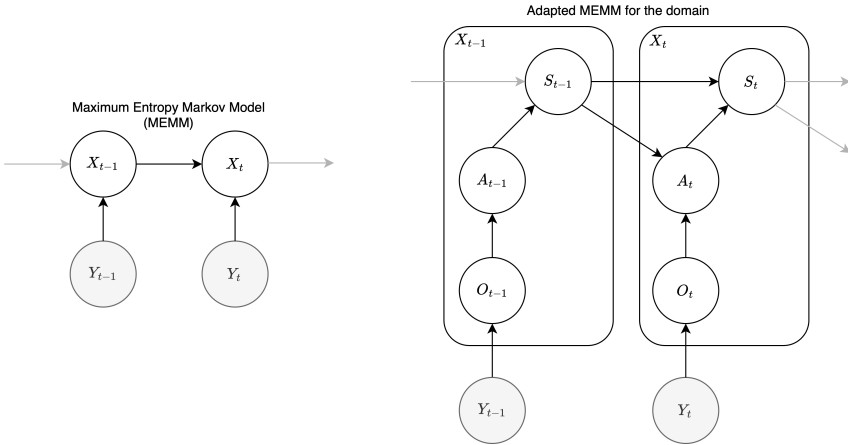

Figure 7: (Left) A standard MEMM, (Right) Our extended version.

Furthermore, we introduce an additional random variable $O_t$ which reflects the symbolic observation we can derive from $y_t$. Given the model in Figure 7, we can factor:

$$P(A_t|S_{t-1}, y_t) = \sum_{O_t} P(A_t, O_t|S_{t-1}, y_t)$$
$$= \sum_{O_t} P(A_t|O_t, S_{t-1}, y_t) P(O_t|S_{t-1}, y_t)$$
$$= \sum_{O_t} P(A_t|O_t, S_{t-1}) P(O_t|y_t)$$

where we assume that $y_t$ is continuous and $O_t$ discrete in nature. Following the example domain given in Subsection 4.1 $y_t$ represents a thermal image and $O_t$ the location of the person for time step $t$.
We now can factor $P(S_T|y_{1:T}, s_0)$ in the following fashion

$$P(S_T|y_{1:T}, s_0) = \sum_{\substack{S_{1:T-1}, \\ A_{1:T}, \\ O_{1:T}}} \prod_{t=1}^{T} P(S_t|S_{t-1}, A_t) P(A_t|O_t, S_{t-1}) P(O_t|y_t)$$

letting DeepProbLog handle the summation over all possible worlds via probabilistic logic inference. Therefore, this approach captures domain constraints (via symbolic rules) while automatically learning needed features (via neural predicates) from data.

## Appendix D. Model architectures

As already mentioned DeepProbLog and the baseline CNN-RNN approach both use a convolutional neural network (CNN). The DeepProbLog CNN processes an 8×8 single-channel input image and is structured as follows: The input image is first processed by a 2D convolutional layer that employs a 3×3 kernel with a padding of 1. This layer extracts a set of feature maps, yielding an output tensor with dimensions 3x8x8 and uses ReLU as an activation function. A subsequent max-pooling operation with a 2x2 kernel and a stride of 2 reduces the spatial resolution to 3x4x4. The reduced tensor is then passed through a second 2D convolutional layer, which is configured identically to the first. This operation expands the feature representation, producing an output of size 6x4x4. A second max-pooling layer is applied to further down-sample the tensor to dimensions 6x2x2. The resulting tensor is flattened into a 24-dimensional vector. This vector is fed into a fully connected (feed-forward) network that comprises a hidden layer with 12 neurons activated by the ReLU function, followed by an output layer of 5 neurons, where the Softmax activation is used to generate class probabilities.

For the baseline CNN-RNN model (which can be seen in Figure 10), the architecture of the CNN slightly differs. As it can be seen in Figure 9 it also uses a 2D convolutional layer that employs a 3×3 kernel with a padding of 1. This layer extracts a set of feature maps, yielding an output tensor with dimensions 16x8x8. On this tensor the CNN applies Batch Normalization, before applying the ReLU function. Afterwards the resulting tensor is processed by a second 2D convolutional layer, which is configured identically to the first, but uses 6 channels, thus resulting the tensor into 32x8x8 shape. This tensor is flattened into a 2048 dimensional vector and reduced to 5 dimensions by a fully connected layer.

The CNN output serves as the input to a recurrent neural network (RNN) layer, which consists of 15 neurons and contains a total of 315 trainable parameters. This RNN is designed to capture temporal dependencies over the sequential data. The final hidden state from the RNN is then transmitted to a linear layer with 15 neurons. Each neuron in this layer corresponds to one of the 15 distinct final states that can be observed after three discrete time steps. This detailed architecture ensures that the network is capable of extracting robust spatial features (e. g. the position of the protagonist) and effectively modeling temporal dynamics.

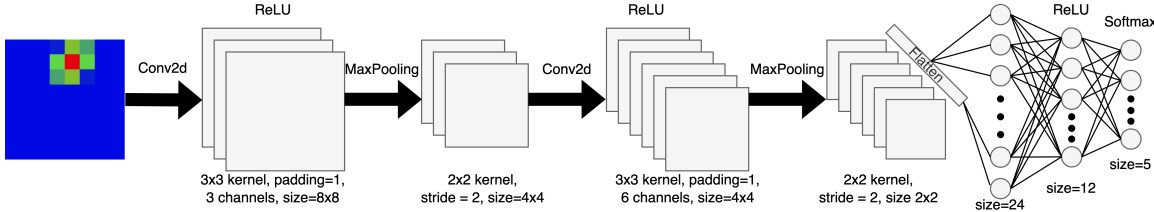

Figure 8: The architecture of the CNN used for the DeepProbLog model.

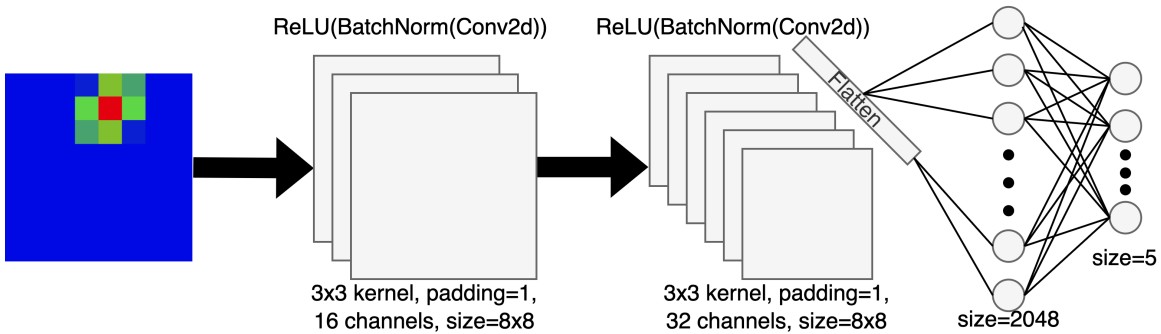

Figure 9: The architecture of the CNN used for the CNN-RNN model.

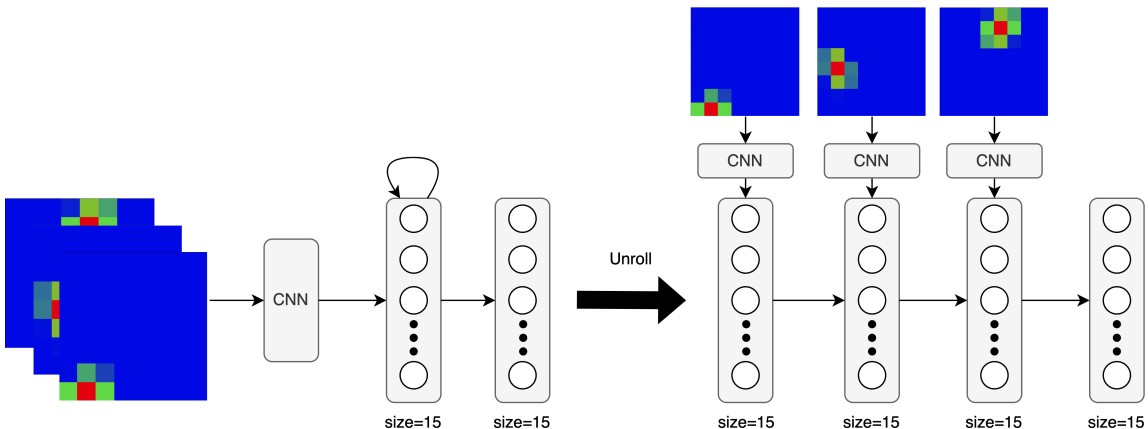

Figure 10: The overall architecture of the CNN-RNN model.

