# OpenReview forum: "Neurosymbolic Learning in Structured Probability Spaces: A Case Study"
_nesyconf.org/NeSy/2025/Conference — NeSy 2025 Poster_

### Official Review · Reviewer_c7jm · 2025-03-25
**An innovative setup for NeSy analysis and future research**

**Rating:** 7
**Confidence:** 3

**Review:**

This paper explores the application of neurosymbolic learning for sequence modeling in structured probability spaces, where learning is challenged by combinatorial explosion and limited supervision. The authors leverage DeepProbLog to integrate symbolic constraints into a neural perception pipeline, and benchmark its performance against a purely neural CNN-RNN model on a simulated human activity recognition task. The study evaluates three key aspects of learning—sample efficiency, generalization, and zero-shot reasoning—through a well-controlled experimental design.

The paper’s main contribution lies in its innovative experimental setup, which provides a clean and novel setting for neurosymbolic research. By simulating a domain with structured constraints, limited observability, and noisy input, the authors effectively highlight the strengths of symbolic integration in scenarios where neural methods typically struggle. The design allows for a fine-grained analysis of the impact of symbolic knowledge under different data regimes. The paper is also clearly written and includes a detailed explanation of the DeepProbLog formalism, making it accessible to a broad audience.

One area for improvement is the limited scope of neurosymbolic techniques considered. While DeepProbLog is a strong representative of logic-integrated models, the paper could be strengthened by discussing or comparing alternative neurosymbolic approaches—for example, methods that relax logical rules into soft constraints (e.g., semantic loss), use differentiable logic layers to capture uncertainty (e.g., SATNet) in symbolic reasoning, or use logic abduction to maximize the consistency between neural networks and symbolic reasoning (e.g., abductive learning). Including such perspectives could help position the work more broadly within the growing field of neurosymbolic AI.

**Anonymity:**

Disclose identity

---

### Official Review · Reviewer_vKw3 · 2025-04-05
**Interesting case study, may generate some discussion during the meeting.**

**Rating:** 6
**Confidence:** 3

**Review:**

The authors present an analysis of tools that combine logic programming and neural machine learning, specifically looking at the DeepProbLog system that offers this combination. The DeepProbLog system has been around for some time, and it has been used in a variety of ways; also it should be noted that using simulated data hurts a bit the conclusions here. Anyway, it is interesting to see this case study based on these ideas, but the value is somewhat incremental as the issues have been examined. In any case, the results may spark debate at the meeting. Overall, the presentation is very good, with clear writing.
-

**Anonymity:**

Remain anonymous

---

### Official Review · Reviewer_1wj7 · 2025-04-09
**The paper provides valuable empirical insights into neurosymbolic learning's effectiveness in structured sequence modeling but lacks sufficient methodological differentiation from recent related frameworks.**

**Rating:** 6
**Confidence:** 4

**Review:**

The paper presents an exploration of neurosymbolic learning (NeSy) for sequence analysis in Structured Probability Spaces (SPS), specifically examining its advantages over purely neural approaches in the context of human activity recognition (HAR). It evaluates DeepProbLog, a neurosymbolic framework integrating symbolic domain knowledge with neural networks. The paper systematically tests three learning capabilities: sample efficiency, generalization, and zero-shot learning.

The paper is clear, well-structured, and comprehensively explains the background and method used, making it accessible and thorough. The key contributions include demonstrating the ability of neurosymbolic methods to significantly enhance sample efficiency and generalization, especially under data scarcity conditions. Furthermore, the presented experiments convincingly showcase the zero-shot learning capabilities of neurosymbolic approaches, an aspect often challenging for purely neural models. Specifically, the paper demonstrates that DeepProbLog maintains robust performance even with substantial reductions in training data or entirely unseen states during training.

Originality:
The paper applies neurosymbolic methods to a structured sequential domain, notably human activity recognition. While it utilizes the well-known DeepProbLog framework without proposing significant architectural innovations, the application scenario is illustrative and provides clear empirical validation of neurosymbolic techniques in structured probability spaces. However, the novelty primarily lies in the empirical evaluation rather than methodological innovations. The paper does not explicitly differentiate itself from closely related frameworks such as DeepSeaProbLog, which handles continuous variables, NeSy-MMs, which specifically target sequential probabilistic reasoning and scalability, or frameworks such as Logic Tensor Networks (LTNs) or Logical Neural Networks (LNNs), which emphasize interpretability and constraint satisfaction. This omission limits a full assessment of the paper’s originality relative to very recent neurosymbolic approaches.

Strengths:

- A clearly articulated problem setup and comprehensive evaluation across relevant scenarios (sample efficiency, generalization, zero-shot learning).
- Strong empirical evidence supports the hypothesis that symbolic constraints effectively constrain the hypothesis space and improve learning efficiency.
- Demonstration of robustness and stability in neurosymbolic models, particularly notable under scenarios of extreme data scarcity.

Comparative Limitations:

- The paper does not sufficiently compare or position its approach relative to the latest developments, such as NeSy-MMs (for handling sequences with relational logic), DeepSeaProbLog (for hybrid discrete-continuous domains), Logic Tensor Networks (LTNs), and Logical Neural Networks (LNNs), all of which are directly relevant and important benchmarks.
- Given the methodological closeness to standard DeepProbLog, the presented work risks partially overlapping with established methods without clearly articulating unique or differentiating aspects.
- The experiments are conducted in a relatively small and simplified domain (125 states, 12 actions), which raises concerns about the general applicability or scalability to larger or more complex real-world scenarios. Addressing or clarifying this explicitly would strengthen the paper’s contributions.

Overall, the paper  requires clearer methodological differentiation and broader comparisons to fully realize its potential impact.

References:

- De Smet, Lennert, et al. "Relational Neurosymbolic Markov Models." arXiv preprint, arXiv:2412.13023, 2024. [https://arxiv.org/abs/2412.13023](https://arxiv.org/abs/2412.13023)
- Badreddine, Samy, et al. "Logic Tensor Networks." Artificial Intelligence, vol. 303, 2022, p. 103649. [https://doi.org/10.1016/j.artint.2021.103649](https://doi.org/10.1016/j.artint.2021.103649)
- Agravante, Don J., et al. "Learning Neuro-Symbolic World Models with Logical Neural Networks." OpenReview preprint, 2023. [https://openreview.net/forum?id=VD0ksRTljb](https://openreview.net/forum?id=VD0ksRTljb)

**Anonymity:**

Remain anonymous